# Differences in Trends in Admissions and Outcomes among Patients from a Secondary Hospital in Madrid during the COVID-19 Pandemic: A Hospital-Based Epidemiological Analysis (2020–2022)

**DOI:** 10.3390/v15071616

**Published:** 2023-07-24

**Authors:** Rafael Garcia-Carretero, Oscar Vazquez-Gomez, María Ordoñez-Garcia, Noelia Garrido-Peño, Ruth Gil-Prieto, Angel Gil-de-Miguel

**Affiliations:** 1Department of Internal Medicine, Mostoles University Hospital, 28935 Móstoles, Madrid, Spain; govaz5@hotmail.com; 2Department of Hematology, Mostoles University Hospital, 28935 Móstoles, Madrid, Spain; maria.ordonez@hotmail.com; 3Department of Pharmacy, Mostoles University Hospital, 28935 Móstoles, Madrid, Spain; noelia.garrido@salud.madrid.org; 4Department of Preventive Medicine and Public Health, Rey Juan Carlos University, 28922 Alcorcón, Madrid, Spain; ruth.gil@urjc.es (R.G.-P.); angel.gil@urjc.es (A.G.-d.-M.)

**Keywords:** COVID-19, health care, SARS-CoV-2, time series visualization

## Abstract

Spain had some of Europe’s highest incidence and mortality rates for coronavirus disease 2019 (COVID-19). This study highlights the impact of the COVID-19 pandemic on daily health care in terms of incidence, critical patients, and mortality. We describe the characteristics and clinical outcomes of patients, comparing variables over the different waves. We performed a descriptive, retrospective study using the historical records of patients hospitalized with COVID-19. We describe demographic characteristics, admissions, and occupancy. Time series allowed us to visualize and analyze trends and patterns, and identify several waves during the 27-month period. A total of 3315 patients had been hospitalized with confirmed COVID-19. One-third of these patients were hospitalized during the first weeks of the pandemic. We observed that 4.6% of all hospitalizations had been admitted to the intensive care unit, and we identified a mortality rate of 9.4% among hospitalized patients. Arithmetic- and semi-logarithmic-scale charts showed how admissions and deaths rose sharply during the first weeks, increasing by 10 every few days. We described a single hospital’s response and experiences during the pandemic. This research highlights certain demographic profiles in a population and emphasizes the importance of identifying waves when performing research on COVID-19. Our results can extend the analysis of the impact of COVID-19 and can be applied in other contexts, and can be considered when further analyzing the clinical, epidemiological, or demographic characteristics of populations with COVID-19. Our findings suggest that the pandemic should be analyzed not as a whole but rather in different waves.

## 1. Introduction

Severe acute respiratory syndrome coronavirus 2 (SARS-CoV-2) is the coronavirus responsible for the coronavirus disease 2019 (COVID-19) pandemic and its more serious consequence, a severe respiratory illness called SARS [1,2]. It was first identified in the city of Wuhan (Hubei, China). The World Health Organization declared the outbreak a public health emergency on 30 January 2020, and eventually a pandemic on 11 March 2020 [3,4]. SARS-CoV-2 is a single-stranded RNA virus that often affects humans [5]. According to the U.S. Department of Health and Human Services, this virus is related to SARS-CoV-1, which caused an outbreak of SARS between 2002 and 2004 [6,7], and with the Middle East Respiratory Syndrome coronavirus (MERS-CoV), which first occurred in 2012 and has been causing persistent endemics in the countries of the Middle East [8,9].

Outbreaks of pandemics typically spread in regular patterns, usually as logarithmic increases in the number of confirmed cases, which are also called exponential curves. However, waves of COVID-19 have varied widely among countries and even regions within a single country depending on the intensity of government and public measures and interventions, along with other factors such as the use of lockdowns, social distancing measures, vaccination, or border policies [10,11]. For example, there were several differences in incidence and mortality rates between countries such as Spain or Italy (with mortality rates up to 15%) and countries such as Germany or Canada (with mortality rates less than 5%) in the first wave of COVID-19 [12,13,14]. The reasons for these differences remain unclear, although some authors have proposed differences in characteristics of the population, government strategies, heterogeneous health systems, or public health interventions [15,16].

Several drugs have been introduced for the treatment of COVID-19. Corticosteroids, such as dexamethasone, have demonstrated efficacy in reducing inflammation and improving outcomes in severe cases. Plasmapheresis, a procedure that removes and replaces blood plasma, has been explored as a potential treatment option to remove harmful antibodies in critically ill patients. Anticoagulants, such as heparin, are administered to prevent blood clotting complications associated with COVID-19. Immunomodulators, such as tocilizumab, act to regulate the immune response and are utilized in severe cases with cytokine release syndrome. Antiviral drugs, including remdesivir, target the replication of the SARS-CoV-2 virus. These drugs, used in various combinations and based on disease severity, have shown promise in improving outcomes and reducing the severity of COVID-19 [17,18,19,20,21,22,23,24].

### 1.1. The Epidemiological Situation in Spain: A Timeline

Clinical and demographic data on the first wave of COVID-19 in Spain were published early on and offered an overall view of the pandemic [25]. Spain’s public health system and intensive care units (ICUs) were overwhelmed with the excessive workload, the high incidence of hospital admissions, and deaths due to COVID-19 [26]. Hospitalizations due to common illnesses and programmed, noncritical surgical interventions decreased. Although some research has been performed in Spain regarding demographic characteristics [25,27,28], to the best of our knowledge no study has compared clinical data and outcomes for different waves. It is key to highlight the importance of identifying waves when performing research on COVID-19 at a given time and in a given location to assess evidence-based decision-making and the impact of COVID-19.

The first case of COVID-19 in Spain was confirmed on 31 January 2020. When the World Health Organization declared the existence of a pandemic on 11 March 2020, Italy and Spain had the highest incidence in Europe, and Spain declared a state of emergency on 14 March 2020. In addition to the lockdown, several measures were instituted, such as mobility restrictions, border closings, and mandatory masking [29]. Spain had one of the highest incidences of COVID-19 in Europe, accounting for 172,541 confirmed cases and 18,056 deaths in the first wave (14 April 2020) [30,31,32]. As of 16 June 2023, a total of 3,905,048 confirmed cases and 121,622 deaths had been reported in Spain [31]. The global incidence was 767,984,989 confirmed cases and 6,943,390 deaths.

### 1.2. The Importance of Data Visualization

Understanding the global pandemic of COVID-19, with its vast amount of data and statistics, can be overwhelming. Data visualization plays a crucial role in identifying trends and gaining insights into the pandemic. However, it is important to note that not all statistics are reliable, and the way data are presented can influence our perception of the situation. Therefore, researchers need to effectively display and represent data to comprehend the outbreak better; although a simple daily count of new cases is the easiest way to present pandemic data, it can be misleading without proper context. To grasp the evolving nature of the pandemic, graphs like histograms, scatterplots, and line plots provide more meaningful trends at a glance. Additionally, a cumulative graph, which shows the total number of confirmed cases per day since the beginning of the pandemic, can be helpful. However, it is essential to exercise caution with cumulative graphs, as they might not clearly indicate if the growth rate is slowing. Researchers must identify a plateau in the curve to demonstrate a slowdown, as cumulative charts always show increasing cases [33].

With arithmetic-scale graphs, researchers can easily identify patterns or trends. The distance along any axis always represents the same quantity. In our research, the space between tick marks along the y-axis (vertical axis) is the same, as the y-axis shows a continuous variable (admissions, cases, deaths). As a result, the distance from 1 to 10 is the same as the distance from 11 to 20. Ticks represent absolute values. If the same data were displayed using a logarithmic scale for the y-axis, we would obtain a semi-logarithmic-scale line graph. In this chart, the distance from 1 to 10 is the same as the distance from 10 to 100. This means that the y-axis is ranked in order of magnitude (10${}^{0}$, 10${}^{1}$, 10${}^{2}$, 10${}^{3}$). We use a semi-logarithmic scale in certain cases that are especially useful for understanding the impact of the pandemic, in particular if the disease is growing exponentially.

### 1.3. Objectives of This Research Study

Improved knowledge of the distribution of confirmed cases, admissions, and deaths due to COVID-19 throughout subsequent phases of the pandemic shed light on the behavior of the virus and its impact on the health care system. New insights will help public health authorities make appropriate decisions and design interventions to manage the pandemic. In light of recent events, our research endeavors to utilize the most up-to-date data from our hospital to examine and investigate the implications and repercussions of the pandemic. Our primary objective is to elucidate the temporal progression of various key variables, such as hospitalizations, occupancy rates, ICU admissions, and deaths, in order to uncover any discernible patterns and trends that may be underlying. Additionally, by presenting these data, we aim to quantify the extent of the health care impact caused by COVID-19 within a specific hospital setting. This valuable information has the potential to contribute to the overall understanding of individual hospital experiences and even serve as a reference for health care systems on a national scale, aiding in their assessment of the pandemic’s effect on their own systems.

## 2. Materials and Methods

### 2.1. Data Collection

We conducted a retrospective, descriptive, epidemiological study in which we determined the frequency and distribution of cases of the COVID-19 pandemic. We included individuals admitted to the hospital whose cause of hospitalization was COVID-19. Therefore, hospitalization due to COVID-19 was defined as having a confirmed infection with SARS-CoV-2 (usually, a positive polymerase chain reaction (PCR) test result). Patients whose admission criteria or discharge report included severe acute respiratory infection as the cause of hospitalization were included. We analyzed trends in newly confirmed cases admitted to our hospital, occupancy time, and mortality rates. Because this study was merely a descriptive investigation, no hypotheses were made. Data were collected from electronic records of Mostoles University Hospital (Spain). Age, sex, admission, discharge dates, comorbidities, drug therapy, status at discharge (alive or dead), and ICU admission were collected. Hospital and ICU stay refer to the length of stay; that is, the duration (in days) of a single episode of hospitalization or ICU admission. Furthermore, by incorporating sex-disaggregated data into statistical presentations, we promote a more inclusive and accurate understanding of the clinical presentation of COVID-19, which would allow us to uncover gender-based patterns, disparities, and trends that may remain hidden in aggregated data. Each patient was given a unique identification number to ensure anonymity. This study was approved by the Ethical Board of Mostoles University Hospital (CEIC 2020/025). According to official sources, our hospital attends to a population of 168,000, with more than 500 hospital beds and 12 critical care beds available [34]. Mostoles University Hospital can be considered a secondary hospital (also known as an intermediate complexity hospital).

Given that the COVID-19 pandemic is still ongoing, we established a research window from the beginning of the pandemic in our hospital (25 February 2020) to the end of the observation period on 12 May 2022. This observation window covered six waves (almost 27 months) of the pandemic. We are aware that every country, and even every region within a single country, has had different waves of COVID-19, and the distribution of confirmed cases varies with the implementation of control strategies. Given these regional differences, we describe the experiences and the distribution of the pandemic at our own institution; our splitting of the pandemic into six waves is utterly idiosyncratic and cannot be extrapolated to other settings.

### 2.2. Statistical Analyses

We plotted continuous variables (age, hospital stay, ICU stay) to check for normality. However, because visual inspection can be unreliable, we used the Shapiro–Wilk test. That is, we combined visual inspection and significance testing to ensure that the assumptions of the statistical tests were met. We also performed several tests of independence. Continuous variables were tested with the Mann–Whitney–Wilcoxon test as a nonparametric alternative to the one-sample t-test when the data could not be assumed to be normally distributed. Data were then expressed as means and standard deviations or as medians and interquartile ranges. We also used the one-proportion Z-test with Yates continuity correction to compare observed proportions of patients in each wave, given that there were only two categories (men and women), to determine whether the proportion of men with COVID-19 differed significantly from the proportion of women with the disease. Categorical data (deaths, ICU admissions) were tested with the chi-square test. Log-linear analysis was used to examine the relationship between more than two categorical variables, such as comparing age and sex throughout the different waves. We used this technique exclusively for hypothesis testing (i.e., as a test of independence). Although we could have used Pearson’s chi-square test instead of log-linear analysis, chi-square only allows for a two-way contingency table analysis (i.e., only two variables can be compared at a time) [35,36]. In contrast, log-linear analysis is a form of categorical data analysis used mostly with three-way contingency tables and can be considered an extension of Poisson regression. This is why they are called Poisson log-linear models.

We set the significance level at *p* = 0.05. We used R language (version 4.2.0) and Python (version 3.7.3 with scikit-learn libraries). The use of either language was not exclusive but rather complementary, depending on the ease of producing statistical metrics or visualization of the data.

### 2.3. Data Visualization

As noted previously, the importance of data representation and visualization for exploratory analyses should not be overlooked because the way in which data are represented affects how researchers interpret them and thus what conclusions are drawn from them. Most research focuses on the visualization of data as time series; that is, groups of observations of a single entity ordered in time. Here, we considered several entities or variables: hospital admissions, ICU and hospital occupancy, and deaths. Observations were conducted daily, and we plotted them with an aim to describe (i.e., we tried to interpret their distribution over time and extract basic useful insights). Because our objective was not to forecast the future, we did not analyze factors such as seasonality, autocorrelation, or stationarity.

Apart from visualizing the data, we also produced tables with useful demographic characteristics, which we split into waves. Waves were set according to a specific signal so that we could extract the position and intensity of multiple peaks and valleys. It is important to note that, as mentioned before, we identified the dates of peaks and valleys for our institution, but we are aware that dates vary from hospital to hospital and even among regions of the same country. We calculated peaks and valleys inside our time series using the function find_peaks from scipy.signal (Python language). We then plotted the resultant figures to show the different segments we considered waves.

The time series were plotted as a continuous line, but given the daily variations, this line turned out to be sharp and rough (i.e., very noisy). We then chose to plot the time series as single points (i.e., a scatterplot) but with a smooth line to identify interesting trends in the data. The simplest method of reading data is to use the moving average (i.e., a model that states that an observation is the mean of a window of past observations). We defined a window to apply the moving average model to smooth the time series and highlight different trends. Whereas a scatterplot represents real data, a moving average line represents trends and different waves. A moving average is less sensitive to abrupt changes, outliers, or missing values and corrects the trend of the time series. It thus smooths fluctuations in the short term. We chose a window of 7 days (i.e., we calculated the mean of the previous seven values). We also used population pyramids for both admissions and deaths. A population pyramid shows the percentage or count of the population by age and sex using two histograms. For other representations of data, we followed Allen et al. [37] and introduced rainclouds, an alternative to bar plots and boxplots, to display probability density plots, raw data points, and boxplots, which show complex, heterogeneous data at a glance.

Finally, as mentioned previously, we used arithmetic-scale cumulative and semi-logarithmic-scale cumulative graphs to better represent the impact of the pandemic. We used some of these charts or a combination of them to obtain a better understanding of the data.

## 3. Results

### 3.1. General Characteristics and Waves

We collected data from 3315 hospitalized patients from 25 February 2020, to 12 May 2022. We identified six waves based on the peaks and valleys in the time series (see Figure 1 showing the calculated peaks and valleys). Based on Figure 1, we plotted Figure A1 in the Appendix A, which splits the time series into six different waves, with plateaus of different lengths between each wave. This method of organization allowed us to group patients in waves and analyze their characteristics. Table 1 summarizes our findings, with clinical characteristics, associated comorbidities, and drug therapy. The number of patients decreased in each wave, except for the sixth wave, in which we found 513 patients. COVID-19 affected more men than women, both globally (55% men vs. 45% women) and in each individual wave. Hypertension, obesity, and type 2 diabetes were the most common comorbidities. We observed a decrease in the frequency of all comorbidities over time, especially in the fourth and fifth waves. The most common drug was dexamethasone, which began to be used at the end of the first wave. Since then, almost all patients were on corticosteroids. Patients on immunomodulatory drugs such as baricitinib, tocilizumab, or anakinra decreased over time. Of note, the standard treatment at the beginning of the pandemic was the combination of lopinariv/ritonavir, hydroxychloroquine, and azithromycin, but they were no longer used after the second wave, as they were replaced by improved drug therapies.

Figure A2 summarizes the evolution of the COVID-19 pandemic during the 27 months since the beginning of the outbreak. We plotted daily admissions, ICU/hospitalization ward occupancy, and deaths over time. The median stay was 7 days (interquartile range: 8). Men tended to stay longer than women, both in the hospitalization ward and ICU (Table 2). Figure A3, panel A shows the distribution of hospitalization stays by wave.

### 3.2. Analyses by Sex and Age

Regarding sex and age, we found that globally women admitted to the hospital because of COVID-19 were older than men, except for those impacted during the third wave. It is worth noting that the average age of the entire population dropped to 47 (interquartile range: 32.8) at the end of the fifth wave. Population pyramids for both admissions and deaths are plotted in Figure 2, respectively. Most hospitalized patients were older than 40 years old. Figure 2 also shows that men predominated in the cohort studied, and that mortality was predominant in men older than 60. However, a closer look at the raincloud computed to visualize those data (Figure A3) shows that the distribution of the age was not normal in any of the waves. In fact, the distribution of age is often multimodal, and utilizing the mean or median as a metric can be misleading. Thus, we disaggregated the data by age, sex, and wave and performed a log-linear analysis, which can be understood as a Poisson regression applied to multiway contingency tables, as mentioned earlier. We found there were no significant differences by wave in those aged 0 to 30 years old. We studied mutual independence among age, wave, and sex and joint independence among these three variables (i.e., interactions among variables during the observation period). Admissions in the population younger than 40 remained steady from the first wave until the sixth wave. We also found that admissions of young patients (<61), regardless of sex, decreased over time. However, an interesting phenomenon occurred in admissions of those 61 to >80 years old: the number of hospitalizations decreased steadily until the fifth wave but increased in the sixth wave, with no differences by the sex of the patient.

### 3.3. ICU Admissions

Panel B in Figure A2 shows the impact of the pandemic on the ICU in our hospital. The horizontal line shows the baseline of 12 critical beds, which were often outnumbered by ICU admissions. This forced hospital authorities to create new spaces for critical beds, such as surgery rooms and postoperative beds. We recorded 154 patients admitted to the ICU; similar to ward admissions, the number of ICU patients decreased steadily until the fifth wave but increased in the sixth wave, with no differences in the sex of the patient. The median ICU stay was 19 days in contrast to 7 days for the median ward stay. Men stayed longer than women in the ICU. The distribution of the median ICU stay was more heterogeneous, often multimodal, and had frequent outliers; a single median likely would not convey the real context of the situation, so we plotted the results in Figure A3, panel B, by means of a raincloud chart.

### 3.4. Mortality

We recorded a global mortality rate of 9.4% among hospitalized patients (310 deaths). The data showed that men were more prone to die than women (197 men vs. 113 women, *p* = 0.002), as seen in Figure 2. The first wave had a higher mortality rate (16.6%) than the other waves for both men and women. The fourth and fifth waves had lower mortality rates. A slight increase could be seen in the sixth wave. It is worth noting that we observed this increase globally. Table 3 shows the distribution of mortality by groups of age and sex. Men had a higher mortality rate than women, but when we split the data into waves, we could not find differences in terms of sex.

We plotted several charts to visualize the impact of mortality on our hospital compared to admissions. Figure A4 shows these graphs. We first plotted the cumulative incidence of admissions and deaths among confirmed cases of COVID-19 on an arithmetic scale (panels A and B in Figure A4). However, the semi-logarithmic-scale plots showed the true impact and burden on the hospital (panels C and D in Figure A4). In the first 2 months, there were approximately 1000 admissions and more than 100 deaths, almost one-third of all admissions and deaths in the observation period. In May 2020, the curves tended to flatten and remain steady until January 2021 (third wave). After a stable plateau, an increased incidence was observed in January 2022 (sixth wave).

We also created some plots to visualize mortality over time compared to all discharges. Figure A5 summarizes the data from Table 2. Up to 25% of all persons discharged in the first month of the pandemic (March 2020) died, which emphasizes the great impact of the pandemic on daily work.

### 3.5. The Pandemic in the Area Attended to by Our Hospital

We plotted the confirmed cases in the area attended to by our hospital (population: 168,000) to put our data into context (Figure A6, online data source: see [38]). Here, our aim was to compare public, official data on confirmed cases among the general population to data on those who had been admitted to our hospital. We found differences in the heights of the peaks of incidence.

Finally, we also plotted the ratio of population in the region of Madrid with a complete vaccination schedule (Figure A7, online data source: see [34]), which raised to 55% by August 2021.

## 4. Discussion

The main objective of our research was to analyze and visualize admissions, occupancy, and mortality due to COVID-19 to evaluate the impact of the pandemic on daily work and pressure on the health care system in a peripheral hospital. The outbreak of COVID-19 overwhelmed our ICUs and the capacity of our hospital. We split the pandemic into waves and analyzed each one separately. The waves turned out to be heterogeneous and dissimilar. Patients in each wave had different epidemiological and demographic profiles. The reasons for these differences remain unclear, but we have some hypotheses: age, sex, social events, new treatments, vaccination, and SARS-CoV-2 variants. Although here we discuss these hypotheses, we can only establish the association, but not causation, between certain events and peaks of hospitalizations in our geographic area.

Regarding age- and sex-specific analyses, women were less prone to have bad outcomes (ICU admission, death), except in the elderly. The interpretation of such differences was challenging, and inconsistent findings can justify a necessity for a more precise analysis that would elucidate the impact of sex and age in the outcomes of COVID-19. Regarding the diagnosis of COVID-19 cases, no apparent sex or gender bias has been observed, although this may vary across countries. However, a noteworthy finding emerges when considering disease progression to severe conditions and mortality, as male individuals exhibit a significant disadvantage. A hypothesis would be that men tend to die earlier than women globally, so it could be that COVID-19 is exacerbating underlying mortality differences. The existence of biological differences in immune systems between men and women can influence their ability to combat infections, including SARS-CoV-2. Generally, females exhibit greater resistance to infections compared to males. Furthermore, lifestyle choices, such as higher rates of smoking and alcohol consumption among men, may contribute to this disparity. Moreover, it is noteworthy that women tend to display a more responsible attitude towards the COVID-19 pandemic when compared to men [39,40].

The use of new drugs, beginning in summer of 2020, can explain the lower mortality after the first wave, shown in Table 1 and Figure A4. This can be explained by the use of new treatments, such as corticosteroids [41,42], antivirals, and immunomodulatory drugs, such as dexamethasone, remdesivir, anakinra, tocilizumab, or baricitinib [19,20,21,22,23,24]. Corticosteroids are beneficial in treating severe COVID-19 cases by reducing lung inflammation and preventing complications. Dexamethasone, studied in the RECOVERY trial, has been shown to reduce mortality in hospitalized patients requiring supplemental oxygen or mechanical ventilation. Remdesivir can shorten recovery time, especially in severe cases, but its impact on mortality reduction remains inconclusive. Combining the immunomodulatory drug baricitinib with remdesivir has demonstrated faster recovery and improved outcomes in hospitalized patients, particularly those needing supplemental oxygen or high-flow therapy. Regarding ventilation, the choice between invasive and non-invasive ventilation plays a critical role in the outcome of patients with COVID-19 in ICUs. It is typically employed in patients with severe respiratory failure or acute respiratory distress syndrome (ARDS). In contrast, non-invasive ventilation provides respiratory support through a mask or nasal prongs without the need for intubation, and although both ventilation strategies aim to support breathing, invasive ventilation is associated with higher levels of respiratory support and is often used in more critically ill patients. The choice of ventilation mode can significantly impact patient outcomes, with invasive ventilation generally being associated with higher mortality rates compared to non-invasive ventilation. However, the decision regarding the appropriate ventilation strategy should be individualized, taking into account factors such as disease severity, patient characteristics, and careful assessment of risks and benefits.

Our cohort had prevalent conditions such as type 2 diabetes, metabolic syndrome, or cardiovascular disease. These conditions, along with advanced age, have been associated with worse outcomes in individuals infected with SARS-CoV-2. Older patients with pre-existing conditions are particularly vulnerable, as age can weaken the immune system and make individuals more susceptible to severe illness. Moreover, comorbidities can further increase the severity of COVID-19 symptoms, contribute to a higher risk of complications, and lead to a higher mortality rate. These underlying health conditions and the aging process can exacerbate the inflammatory response triggered by the virus, resulting in complications such as acute respiratory distress syndrome and multiorgan dysfunction, which can explain the high mortality among the elderly in our cohort.

The first wave was associated with the initial outbreak and was restrained by strict public health measures, such as confinement and lockdown. The second wave began in the summer of 2020, when those restrictions ended and social distancing measures were relaxed. This wave reached a peak in the autumn of 2020, probably because of the return to work and school. The third peak began in December 2020, probably as a result of holiday events and Christmas gatherings, and continued until January 2021. The next waves showed a rapid fall in hospitalizations, probably because of vaccination, and its peak might have been associated with the Easter holidays. Since then, waves showed the beneficial effect of vaccination. Regardless of vaccination, it seems that waves and peaks were related to social events: holidays, gatherings, and the relaxation of public health measures such as social distancing.

Vaccination began in the European Union (and also in Spain) in December 2020, and its protective effects can explain the lower admissions and lower mortality among the elderly since [43,44]. Beginning in April 2021, there was a rapid decline in admissions due to COVID-19. Although we are aware that interpretation can be challenging, some authors demonstrated the beneficial impact of vaccination [45]. The use of vaccines resulted in a decrease in hospitalizations, probably because their protection was based on achieving a mild clinical presentation of COVID-19. They had a great impact in terms of hospital admissions and mortality. As mentioned, the effect of vaccination since the summer of 2021 was studied by Barandalla et al. [45], who developed simulated curves of hospitalizations in the absence of vaccines and then compared those curves with the real incidence. By showing the decrease in incidence, they demonstrated the beneficial impact of the vaccination rollout on hospitalizations. With our study, we hypothesize that this impact was steady in the fourth and fifth waves; that is, the steady vaccination of the elderly consolidated the decline in admissions due to COVID-19. The authors state that new COVID-19 hospitalizations occurred in younger, non-vaccinated individuals. They demonstrated that the elderly were not the most frequently hospitalized group since mid-May 2021, but individuals <50 not yet vaccinated.

An interesting point arises regarding vaccination and the subsequent drop in mortality among elderly individuals. Some recent publications have focused on choosing the best vaccination strategy based on certain criteria [46,47]. A protocol should be applied depending on the population structure when the aim is to prevent the spread of an illness, limit the number of deaths, and reduce the impact on health care. In those studies, the authors focus on priority based on population structure (i.e., first vaccinating elderly individuals, who accounted for the most vulnerable group in Spain). They demonstrate that different disease characteristics and different population structures may play an important role in the choice of certain vaccination protocols. In the case of Spain, a country included in one of the publications [46], vaccinating the elderly resulted in a reduction in overall mortality and was probably responsible for the increased number of confirmed cases and admissions among the unvaccinated, younger population.

In addition, changes in SARS-CoV-2 variants can explain the beginnings of several waves and the different patient profiles. It is beyond the scope of this research to provide extensive background on the variants or different viral lineages of SARS-CoV-2 [48,49]. Nevertheless, we can hypothesize that some waves were related to changes in the predominant variant. Several strains were detected in Spain during the first wave [50], probably due to several genetic variants. By September 2020, a new variant (B.1.1.7, also known as the alpha variant) had been described in Europe and was spreading rapidly in several countries [51]. This variant, which had increased transmissibility, virulence, and lethality, may have been responsible for several waves up to summer 2021 (i.e., up to the fifth wave). However, the spread of the alpha variant coincided with the beginning of vaccination in Europe. We hypothesize that, thanks to the vaccines, both admissions and deaths dropped from the second wave to the fifth wave. In August 2021, the delta variant (B.1.617.2) replaced alpha as the predominant variant in Spain [48,52], affecting a younger, unvaccinated population in the fifth wave (summer 2021). This may explain why hospitalization was more frequent among young people, as we mentioned previously (with a median age of 47, according to Table 1; see also Figure A3).

As we also mentioned earlier, changes in variants played a key role in the distribution of the pandemic. The omicron variant and its descendants (B.1.1.529) were identified in November 2021 [53]. This variant was more transmissible than previous variants, and by January 2022 (sixth wave: from November 2021 to March 2022) it was predominant in Spain [46,47]. This variant affected the population even if they were vaccinated because it was more infectious and could evade their immune responses. Specifically, with regard to the sixth wave, we found more admissions and an older hospitalized population than in the fourth and fifth waves. We also found that mortality increased slightly with respect to the previous waves. We hypothesize that this is because of both the omicron variant and the older age of patients. Fortunately, the number of admissions and deaths was lower than in the first wave, which highlights the effectiveness of widespread vaccination [54]. Although it should be properly analyzed in further studies, we hypothesize that the omicron variant was responsible for the sixth wave.

In terms of impact on health care, the semi-logarithmic-scale charts in Figure A4 are worthy of discussion. In the first wave, the initial outbreak phase, SARS-CoV-2 spread exponentially rather than arithmetically, so a log scale is the natural way to track the spread. As we have shown, vertical distances represent multiplicative differences (i.e., 10${}^{0}$, 10${}^{1}$, 10${}^{2}$, 10${}^{3}$, etc.). Cases increased by 10 every few days. The chart emphasizes the growth and progression of the outbreak in the first wave and its impact on health care in our hospital in terms of admissions and mortality. As mentioned earlier, almost one-third of all admissions and deaths in the observation period occurred during the first weeks of the pandemic. This emphasizes the great impact of the pandemic on daily work. The growth is less pronounced over time (i.e., the curve flattens).

The impact of the COVID-19 pandemic on health care systems in Spain has also been studied up to the third wave, before vaccination, in December 2020 [55]. Analyses of the first, second, and third waves revealed several differences, such as a rise in the number of confirmed cases in the general population due to the less restrictive testing policy (the use of rapid antigen tests), a lower number of severe cases requiring admission to the ICU, and decreased mortality rates. The lower mortality and smaller number of patients requiring ICU admission were studied by Taboada et al. [56], who proposed that corticosteroids and new immunomodulatory drugs were responsible for this phenomenon. In international studies, it was also found that demographic and clinical features of patients with confirmed COVID-19 differed between the third wave and previous waves [57,58,59].

### 4.1. Epidemiological Modeling

Regarding the distribution and patterns of the time series, it is possible to rely on hospital admissions to fit predictive models rather than on confirmed cases in the general population. Although the fitting of predictive models is beyond the scope of this research, it is worth noting that analyses of time series based on confirmed cases in the general population may not represent the real state of the pandemic [60]. The number of confirmed cases depends on the testing policy: the more tests performed, the higher the incidence in the general population and the lower the hospitalization rate. For example, according to official data (Figure A6, see data source in [34]), there was a decoupling between the first wave in the area near our hospital and the first wave in terms of admissions, ICU occupancy, and mortality, mainly because of the restrictive testing policy (only patients strongly suspected of having COVID-19 were tested). We hypothesize that the reason why the first wave is decoupled with respect to admissions is because of such a testing strategy. In contrast, it can be observed in Figure A3 that the rest of the lines are coupled from the second to the fifth waves in terms of admissions, occupancy, and mortality, probably because the testing policy was less restrictive [61,62]. A similar phenomenon can be observed in the sixth wave, as a decoupling between Figure A3 and Figure A6 can be seen. There were more than 5000 confirmed cases among our population, but we recorded only 513 hospitalized patients. We hypothesize that a much more permissive testing policy allowed the detection of even mild cases of COVID-19 that did not require hospitalization. We would like to emphasize the role of the testing policy, which probably was a key factor in detecting confirmed cases of COVID-19. In the first waves, the testing policy was very restrictive (not extensive), and the peaks of the time series were decoupled with respect to the curve of the admissions in the same period. In contrast, in the third, fourth, and fifth waves admissions and hospital/ICU occupancy matched, as expected. Finally, the sixth wave in the area nearby showed a higher peak than expected, probably due to extensive testing.

It is worth mentioning an epidemiological study by Red Nacional de Vigilancia Epidemiológica (Epidemiological Surveillance National System (RENAVE in Spanish)) with analyses of data very similar to ours [32]. Confirmed cases, admissions, and deaths were analyzed up to 10 May 2022, with different visualizations, focused specifically on analyses of age ranges. In that research, data from Spain were reported. The authors split the outbreak into periods and established a turning point for each wave based on the 14-day cumulative incidence. The waves in that research are coupled with those in our study.

### 4.2. Public Health Measures

A question that may arise in the future is what relevant public health measures should be taken by countries. The results, the distribution of waves, and the impact of the pandemic analyzed here were based on several variables, such as population structure, the health care system, testing policies, social interventions, non-pharmacological measures, and vaccination strategies. Spain adopted a suppression strategy as an immediate response to the pandemic based on the aforementioned variables. The aim of this strategy was to reduce the spread of the virus and mortality. In contrast, Barat et al. [63] described the interesting approach adopted by Sweden. Instead of a suppression strategy, Sweden chose a mitigation policy based on its own priorities and legal system. This mitigation strategy was applied in the first wave (March–April 2020) and consisted of risk-tailored measures to protect elderly individuals. In choosing a mitigation policy, Swedish authorities were trying to avoid the potential socioeconomic inequities that are often associated with the massive lockdowns seen with suppression strategies.

Regarding outcomes, Sweden made specific recommendations to prioritize the protection of elderly individuals. Seroprevalence among the elderly was lower in Sweden than in Spain [64]. Mortality in Sweden was higher than in other Scandinavian countries but lower than in other European countries [63]. It is beyond the scope of our study to discuss whether countries should have adopted a mitigation strategy (such as Sweden) or a suppressive one (such as Spain), and the ability to make comparisons is limited by affected populations, testing policies, the timeline of the pandemic, and socioeconomic determinants. Other countries, such as Denmark and Norway, pursued a suppression strategy [65]. Outside Europe and North America, public health strategies in six Asian countries were also analyzed [66]. It is the responsibility of national experts to assess the different approaches and dominant ideas on public health measures regarding the pandemic.

Concerning testing policy, a recent publication by Zhang et al. [67] demonstrated that mass testing was associated with 25% cut admissions due to COVID-19. The city of Liverpool (United Kingdom) was selected for a pilot study. The intervention was to test asymptomatic individuals to identify infected people in order to protect vulnerable individuals, to quarantine the contacts, and ultimately to improve public health. This intervention reduced COVID-19-related admissions because promoting effective isolation of confirmed individuals and their contacts resulted in reduced onward transmission. The study estimated a 32% reduction in admissions compared with the expected admissions with no intervention.

Finally, an intriguing study highlights a recurring pattern of panic and neglect in funding pandemic preparedness. Resources tend to increase in the aftermath of crises but subsequently decline. The high economic costs incurred by the COVID-19 pandemic further highlight the urgent need for investment in preparedness. Estimates for the annual funding required for pandemic preparedness vary, but they remain relatively small compared to the projected costs associated with events like COVID-19. Sustainable funding for pandemic preparedness necessitates effective collaboration between global health stakeholders and national health system leaders, as demonstrated by the importance of timely health responses when political commitment is present [68].

### 4.3. Limitations

Our aim was to describe trends and distributions of the pandemic in our hospital and to determine the impact on our health care system. Consequently, our research focused on data on hospitalization, not on the total number of confirmed patients in our region. The hospitalization rate (i.e., the proportion of admissions among confirmed cases) varies over time, among countries, and even among regions within the same country, depending on the testing policy [32,61]. Therefore, we propose that widespread testing would improve estimates of the true admissions rate [60,62]. As mentioned previously, when interpreting the impact of COVID-19 on hospitals, researchers cannot rely solely on confirmed cases, because sometimes it is difficult to know what proportion of the population has been tested. Likewise, it can be difficult to estimate the mortality rate among the infected population. Another limitation is the local design of the study, as we are aware that every country, and even every region within a single country, has had different waves of COVID-19. Therefore, the distribution of confirmed cases varies. Given these regional differences, we have described the experiences and the distribution of the pandemic at our own institution. Our splitting of the pandemic into six waves is utterly idiosyncratic and cannot be extrapolated to other settings. Finally, it is important to note that this study is limited by the absence of socioeconomic information, which could have provided valuable insights into the potential influence of socioeconomic factors on disease severity, access to health care resources, and treatment outcomes. In addition, we were unable to examine the impact of physical activity as a potential factor influencing the clinical progression of COVID-19. However, an intriguing study involving 131 individuals demonstrated that patients with sufficient or high levels of physical activity were more likely to experience recovery, whereas those with insufficient activity had an increased risk of death. This suggests a correlation between physical activity and a less severe course of the disease [69].

Some previous publications have conducted studies on the effects of COVID-19, focusing on specific secondary hospitals or limited time periods. These studies provided information about the demographic characteristics and outcomes of the patients included in their respective cohorts [70,71,72,73]. Although we recognize that there may be regional variations in the characteristics of COVID-19, and we acknowledge that our findings may not directly apply to other settings, it is important to note that our study encompassed a period of 27 months during the pandemic and involved over 3000 hospitalized patients. However, the strength of this study lies in its meticulous emphasis on complete follow-up for all patients throughout their entire hospital stay which enhances the reliability and validity of the results. This comprehensive monitoring allows for a detailed understanding of the disease progression, treatment outcomes, and potential complications, providing valuable insights for medical practitioners and researchers alike. This approach ensures a representative sample, minimizing potential biases and increasing the generalization of the findings.

## 5. Conclusions

In this study, our objective was to present a comprehensive overview of trends and patterns observed during various stages of the COVID-19 pandemic. We aimed to highlight differences in demographic data, clinical information, and healthcare indicators. Throughout the course of the pandemic, we witnessed advancements in patient management, such as the development of new drugs and the rollout of vaccination programs. However, we also observed the emergence of different variants and lineages, which ultimately had a significant impact on hospitalization rates and mortality trends. When formulating nonmedical strategies to address future waves of COVID-19 or outbreaks of new infectious diseases, it is crucial to consider these factors, in addition to the clinical experience gained during the pandemic. To gain insights into the diverse patient profiles, we analyzed administrative and demographic data across different waves of the pandemic. By dividing the pandemic into distinct waves, we were able to identify variations in patient demographics, likely influenced by key milestones preceding each wave. These milestones included factors such as the initiation of vaccination programs, changes in SARS-CoV-2 variants, and the implementation of social distancing measures. Our findings suggest that analyzing the pandemic as a whole may not capture the complete picture, and it is more informative to examine individual waves. Factors specific to each wave, such as the timing of COVID-19 infections, vaccination rates, and predominant virus variants, should be considered when designing future clinical studies on the pandemic. As a result, our research provides a general analytical framework that can be applied to other settings.

## Figures and Tables

**Figure 1 viruses-15-01616-f001:**
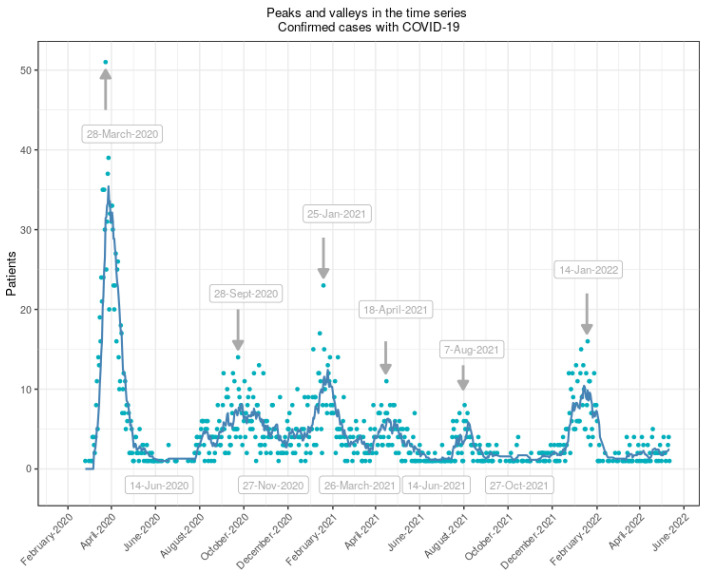
Calculated peaks and valleys inside the time series. Dots represent raw data, whereas the blue line represents a moving average of the time series.

**Figure 2 viruses-15-01616-f002:**
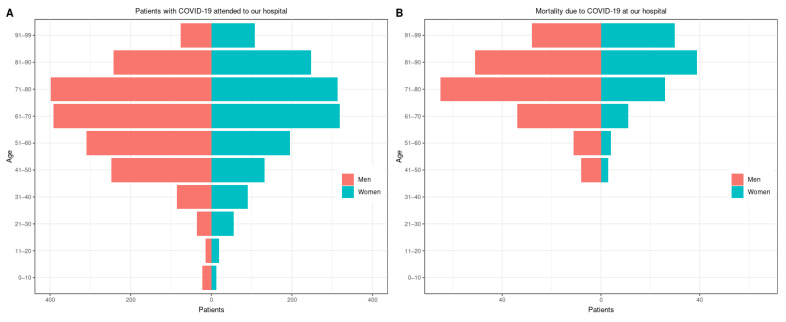
Distributions of patients in the population admitted to the hospital (**A**) and mortality (**B**), both in age–sex pyramids. The plots were computed to display the distribution of the population attended to in our hospital affected with COVID-19. The pyramids depict the impact of the illness by age and sex on admission (**A**) and mortality (**B**).

**Table 1 viruses-15-01616-t001:** Epidemiological and demographic characteristics of patients admitted to our hospital between 2020 and 2022.

	First Wave	Second Wave	Third Wave	Fourth Wave	Fifth Wave	Sixth Wave	All Waves
Patients, n (%)	1024 (30.9%)	652 (19.7%)	646 (19.4%)	278 (8.3%)	202 (6.2%)	513 (15.5%)	3315
Sex							
Men	553	363	359	164	105	279	1823 (55%)
Women	471	289	287	114	97	234	1492 (45%)
Age, median (IQR)	70 (22.2)	65 (26)	66 (23)	60 (21)	47 (32.8)	70 (26)	67 (25)
Age ranges							
<20	3	8	10	4	12	31	68
21–40	43	68	48	24	51	32	266
41–60	246	181	186	112	64	96	885
61–80	476	273	284	119	47	223	1422
>80	256	122	118	19	28	131	674
Comorbidities							
Type 2 diabetes	19.3%	21.8%	21.7%	16.4%	18.4%	21.2%	20%
Hypertension	33.7%	33.2%	34.7%	29.7%	25.5%	32.4%	32.7%
Obesity	8.4%	0.13%	13.6%	15.6%	14.1%	12.4%	11.8%
AMI	6.8%	0.07%	7.5%	4.8%	0.06%	6.6%	6.7%
CHF	6.1%	0.08%	0.08%	3.8%	7.8%	7.3%	6.9%
Dementia	5.2%	4.6%	4.4%	1.9%	4.6%	4.3%	4.5%
Kidney disease	8.9%	9.4%	0.1%	5.3%	9.4%	0.1%	8.9%
Liver disease	0.5%	0.5%	0.5%	0.4%	0.4%	0.6%	0.5%
Malignancy	5.3%	5.9%	0.06%	3.9%	5.6%	7.5%	5.5%
COPD	7.1%	7.3%	8.1%	0.06%	7.4%	8.9%	7.3%
CEVD	0.7%	0.8%	0.8%	0.5%	0.7%	0.01%	0.7%
Drug therapy, n (%)							
Dexamethasone	131 (12.8%)	614 (94.2%)	641 (99.2%)	275 (98.9%)	200 (99%)	502 (97.9%)	2763 (83.3%)
Remdesivir	0 (0%)	230 (35.3%)	156 (24.1%)	96 (34.5%)	41 (20.3%)	79 (15.4%)	602 (18.2%)
Baricitinib	44 (4.3%)	95 (14.6 %)	130 (20.1%)	63 (22.7%)	39 (19.3%)	46 (9%)	417 (12.6%)
Tocilizumab	137 (13.4%)	222 (34%)	90 (13.9%)	61 (21.9%)	21 (10.4%)	27 (5.3%)	558 (4.1%)
Anakinra	12 (1.2%)	19 (2.9%)	13 (2%)	1 (0.4%)	0 (0%)	3 (0.6%)	48 (1.4%)
LPV/r, HCQ, AZM	830 (81.1%)	57 (8.7%)	0 (0%	0 (0%)	0 (0%)	0 (0%)	887 (26.8%)

AMI: acute myocardial infarction. CHF: congestive heart failure. CEVD: cerebrovascular disease. COPD: chronic obstructive pulmonary disease. IQR: interquartile range. LPV/r: lopinariv/ritonavir. HCQ: hydroxychloroquine. AZM: azithromycin.

**Table 2 viruses-15-01616-t002:** Outcomes in terms of ICU admissions and mortality of patients admitted to our hospital (2020–2022).

	Total Patients	Men	Women	*p* Value
Age	67.0 (25.0)	66.0 (24.0)	68.0 (25.0)	0.001 *
First wave	70.0 (22.2)	68.0 (20.0)	72.0 (24.0)	0.001
Second wave	65.0 (26.0)	64.0 (24.0)	67.0 (27.0)	0.159
Third wave	66.0 (23.0)	65.0 (25.0)	68.0 (21.5)	0.004
Fourth wave	60.0 (21.0)	59.0 (20.0)	65.5 (22.8)	0.211
Fifth wave	47.0 (32.8)	47.0 (32.0)	47.0 (34.0)	0.566
Sixth wave	70.0 (26.0)	72.0 (25.5)	69.0 (26.0)	0.2
Hospital stay (days)	7.0 (8.0)	7.0 (8.0)	6.0 (7.0)	0.001 *
First wave	8.0 (9.0)	8.0 (10.0)	7.0 (8.0)	0.103
Second wave	7.0 (9.0)	7.0 (8.0)	7.0 (8.0)	0.068
Third wave	6.0 (7.0)	6.0 (8.0)	6.0 (5.0)	0.028
Fourth wave	7.0 (7.8)	8.0 (7.2)	7.0 (7.8)	0.586
Fifth wave	5.0 (6.0)	5.0 (6.0)	5.0 (5.0)	0.353
Sixth wave	5.0 (6.0)	5.0 (6.0)	5.0 (6.0)	0.588
ICU admissions	154 (4.6%)	108 (5.9%)	46 (3.1%)	0.001 **
First wave	59 (5.8%)	46 (8.3%)	13 (2.8%)	0.001
Second wave	27 (4.1%)	16 (4.4%)	11 (3.8%)	0.853
Third wave	27 (4.2%)	18 (5.0%)	9 (3.1%)	0.323
Fourth wave	17 (6.1%)	11 (6.7%)	6 (5.3%)	0.81
Fifth wave	8 (4.0%)	7 (6.7%)	1 (1.0%)	0.067
Sixth wave	16 (3.1%)	10 (3.6%)	6 (2.6%)	0.614
ICU stay (days)	19.0 (27.0)	18.0 (24.5)	21.5 (35.5)	0.492 *
First wave	7.0 (6.8)	7.0 (7.2)	2.0 (5.0)	0.023
Second wave	6.0 (4.0)	7.0 (4.0)	8.0 (4.8)	0.347
Third wave	5.0 (5.0)	5.0 (5.0)	5.0 (5.8)	0.998
Fourth wave	7.0 (6.0)	10.5 (5.5)	8.5 (10.2)	0.263
Fifth wave	5.0 (5.0)	14.0 (4.8)	49 (5.8)	0.001
Sixth wave	5.0 (5.0)	15.5 (5.0)	17.5 (5.8)	0.625
Deaths	310 (9.4%)	197 (10.8%)	113 (7.6%)	0.002 **
First wave	170 (16.6%)	108 (19.5%)	62 (13.2%)	0.008
Second wave	40 (6.1%)	27 (7.4%)	13 (4.5%)	0.165
Third wave	53 (8.2%)	33 (9.2%)	20 (6.9%)	0.379
Fourth wave	12 (4.3%)	9 (5.5%)	3 (2.6%)	0.37
Fifth wave	8 (4.0%)	4 (3.8%)	4 (4.1%)	1
Sixth wave	27 (5.3%)	16 (5.7%)	11 (4.7%)	0.746

*: Mann–Whitney–Wilcoxon test. **: Chi-square test. ICU: Intensive Care Unit. Hospital and ICU stay refer to the length of stay; that is, the duration (in days) of a single episode of hospitalization or ICU admission. Age and hospital stay are expressed as the median (interquartile range). The rest of the variables are expressed as frequencies and percentages. Percentages regarding ICU admissions and deaths refer to the ICU admission rate and mortality rate, respectively, in each wave.

**Table 3 viruses-15-01616-t003:** Mortality (in absolute values) due to COVID-19 according to age range and sex.

Age Group	1st Wave	2nd Wave	3rd Wave	4th Wave	5th Wave	6th Wave
Men						
<20	0	0	0	0	0	0
21–40	0	0	0	0	0	0
41–60	11	1	2	1	2	2
61–80	58	11	17	7	1	5
>80	39	15	14	1	1	9
Women						
<20	0	0	0	0	0	0
21–40	0	0	0	0	0	0
41–60	5	0	0	0	0	2
61–80	19	4	5	3	1	5
>80	38	9	15	0	3	4
Total						
<20	0	0	0	0	0	0
21–40	0	0	0	0	0	0
41–60	16	1	2	1	2	4
61–80	77	15	22	10	2	10
>80	77	24	29	1	4	13

## Data Availability

A contract signed with Mostoles University Hospital, which provided the dataset, prohibits the authors from providing their data to any other researcher. Furthermore, the authors must destroy the data upon the conclusion of their investigation. The data cannot be uploaded to any public repository.

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
