# Peer review of "Differences in Trends in Admissions and Outcomes among Patients from a Secondary Hospital in Madrid during the COVID-19 Pandemic: A Hospital-Based Epidemiological Analysis (2020–2022)"

_viruses, 2023, doi:10.3390/v15071616_

Round 1
Reviewer 1 Report
I have received for review an original research article entitled “Differences In Trends In Admissions And Outcomes Among Patients From A Secondary Hospital In Madrid During The COVID-19 Pandemic: A Hospital-based Epidemiological Analysis (2020-2022)” prepared by Rafael Garcia-Carretero et al., which is being processed for publication in the journal Viruses (IF=4.7). Although it seems that the greatest epidemic threat related to the COVID-19 pandemic has already passed, a number of cases, including severe ones, are still being recorded. In addition, having several years of experience in caring for patients infected with the SARS-CoV-2 virus, the world of science rightly focuses on making some summaries and analyzing the available data, which can deepen knowledge on this subject. The topic discussed by the Authors of the paper is therefore important. The paper is interesting and very well prepared. However, in my opinion some corrections are necessary that may contribute to further improving the quality and attractiveness of the presented manuscript. I present my suggestions below.
1) The purpose of the work should be clearly defined at the end of the introduction.
2) I believe that it is worth paying a little more attention to the fact that physical activity is also one of the factors that affect the course of SARS-CoV-2 coronavirus infection. It is worth quoting the results of a recent study published in the Journal of Clinical Medicine (10.3390/jcm12124046).
English is understandable in my opinion. I didn't see any serious bugs. However, it is valuable for the text to be checked by a specialist in the field of English philology.
Author Response
Comment #1. The purpose of the work should be clearly defined at the end of the introduction.
Response
The aim of our research was just before the headline “1.1 The epidemiological situation in Spain: a timeline.” However, we agree with the reviewer, since the objectives should be at the end of the Introduction section. Otherwise the reader can overlook them. We have moved a whole paragraph to a more proper location (as suggested by the reviewer).
Comment #2. I believe that it is worth paying a little more attention to the fact that physical activity is also one of the factors that affect the course of SARS-CoV-2 coronavirus infection. It is worth quoting the results of a recent study published in the Journal of Clinical Medicine (10.3390/jcm12124046).
Response
We added the following sentences:
“Moreover, we could not assess physical activity as a modifying factor of the clinical course of COVID-19. An interesting study including 131 individuals stated that patients with sufficient or high physical activity levels were more likely to recover, while those with insufficient activity had a higher risk of death, i.e., physical activity is correlated with a milder course of COVID-19.”
Reviewer 2 Report
This manuscript presents a detailed and well-executed retrospective epidemiological study on the frequency and distribution of COVID-19 cases among individuals admitted to Mostoles University Hospital in Spain. The authors aimed to analyze trends in newly confirmed cases, occupancy time, and mortality rates to provide valuable insights into the impact of the pandemic on the hospital and its patients.
The authors followed a rigorous data collection process, utilizing electronic records from Mostoles University Hospital, ensuring the inclusion of relevant variables such as age, sex, admission and discharge dates, comorbidities, drug therapy, status at discharge, and ICU admission. The use of unique identification numbers to ensure anonymity demonstrates a commitment to ethical considerations.
One commendable aspect of this study is the incorporation of sex-disaggregated data in the statistical presentations. By doing so, the authors promote a more inclusive and accurate understanding of the clinical presentation of COVID-19, allowing for the identification of gender-based patterns, disparities, and trends that may otherwise remain hidden in aggregated data.
The study window, covering six waves of the COVID-19 pandemic from February 2020 to May 2022, provides a comprehensive and in-depth analysis of the evolving situation. The authors acknowledge the regional differences in COVID-19 waves and make it clear that their findings may not be directly extrapolated to other settings, enhancing the transparency and reliability of their results.
The statistical analyses conducted in this study are robust and appropriate for the research objectives. The combination of visual inspection, significance testing, and various statistical tests ensures the validity of the assumptions made. The use of both R language and Python for statistical metrics and data visualization demonstrates the authors' thoroughness in presenting and analyzing the data.
The data visualization techniques employed by the authors effectively present the information and enable a clear understanding of the trends. The use of scatterplots, population pyramids, rainclouds, and moving average lines provides a comprehensive view of the data, highlighting interesting patterns and facilitating interpretation.
The manuscript's results section presents the findings in a systematic and concise manner, starting with general characteristics and waves, followed by analyses of sex and age, ICU admissions, and mortality rates. The tables and figures accompanying the results further enhance the understanding of the presented data.
Overall, this manuscript makes a significant contribution to the field of COVID-19 epidemiology, particularly in the context of hospitalizations. The comprehensive data collection, robust statistical analyses, and effective data visualization techniques contribute to the credibility and reliability of the study. However, I think it is worthwhile for the authors to include some of the articles indicated below in the discussion:
doi: 10.14423/SMJ.0000000000001245
doi: 10.1371/journal.pone.0272953
doi: 10.1093/trstmh/trac039.
doi: 10.5603/CJ.a2021.0043
doi: 10.4081/monaldi.2022.2194
doi: 10.5603/CJ.a2021.0168
This manuscript presents a detailed and well-executed retrospective epidemiological study on the frequency and distribution of COVID-19 cases among individuals admitted to Mostoles University Hospital in Spain. The authors aimed to analyze trends in newly confirmed cases, occupancy time, and mortality rates to provide valuable insights into the impact of the pandemic on the hospital and its patients.
The authors followed a rigorous data collection process, utilizing electronic records from Mostoles University Hospital, ensuring the inclusion of relevant variables such as age, sex, admission and discharge dates, comorbidities, drug therapy, status at discharge, and ICU admission. The use of unique identification numbers to ensure anonymity demonstrates a commitment to ethical considerations.
One commendable aspect of this study is the incorporation of sex-disaggregated data in the statistical presentations. By doing so, the authors promote a more inclusive and accurate understanding of the clinical presentation of COVID-19, allowing for the identification of gender-based patterns, disparities, and trends that may otherwise remain hidden in aggregated data.
The study window, covering six waves of the COVID-19 pandemic from February 2020 to May 2022, provides a comprehensive and in-depth analysis of the evolving situation. The authors acknowledge the regional differences in COVID-19 waves and make it clear that their findings may not be directly extrapolated to other settings, enhancing the transparency and reliability of their results.
The statistical analyses conducted in this study are robust and appropriate for the research objectives. The combination of visual inspection, significance testing, and various statistical tests ensures the validity of the assumptions made. The use of both R language and Python for statistical metrics and data visualization demonstrates the authors' thoroughness in presenting and analyzing the data.
The data visualization techniques employed by the authors effectively present the information and enable a clear understanding of the trends. The use of scatterplots, population pyramids, rainclouds, and moving average lines provides a comprehensive view of the data, highlighting interesting patterns and facilitating interpretation.
The manuscript's results section presents the findings in a systematic and concise manner, starting with general characteristics and waves, followed by analyses of sex and age, ICU admissions, and mortality rates. The tables and figures accompanying the results further enhance the understanding of the presented data.
Overall, this manuscript makes a significant contribution to the field of COVID-19 epidemiology, particularly in the context of hospitalizations. The comprehensive data collection, robust statistical analyses, and effective data visualization techniques contribute to the credibility and reliability of the study. However, I think it is worthwhile for the authors to include some of the articles indicated below in the discussion:
doi: 10.14423/SMJ.0000000000001245
doi: 10.1371/journal.pone.0272953
doi: 10.1093/trstmh/trac039.
doi: 10.5603/CJ.a2021.0043
doi: 10.4081/monaldi.2022.2194
doi: 10.5603/CJ.a2021.0168
Author Response
Overall, this manuscript makes a significant contribution to the field of COVID-19 epidemiology, particularly in the context of hospitalizations. The comprehensive data collection, robust statistical analyses, and effective data visualization techniques contribute to the credibility and reliability of the study. However, I think it is worthwhile for the authors to include some of the articles indicated below in the discussion:
doi: 10.14423/SMJ.0000000000001245
doi: 10.1371/journal.pone.0272953
doi: 10.1093/trstmh/trac039.
doi: 10.5603/CJ.a2021.0043
doi: 10.4081/monaldi.2022.2194
doi: 10.5603/CJ.a2021.0168
Response
We have added the following paragraph to the Discussion section, and included 4 out of 6 proposed references:
“Some previous publications have conducted studies on the effects of COVID-19, focusing on specific secondary hospitals or limited time periods. These studies provided information about the demographic characteristics and outcomes of the patients included in their respective cohorts. While we recognize that there may be regional variations in the characteristics of COVID-19, and we acknowledge that our findings may not directly apply to other settings, it is important to note that our study encompassed a period of 27 months during the pandemic and involved over 3,000 hospitalized patients.”
Reviewer 3 Report
Abstract: Add more results in abstract. Line 13-18 can be deleted and replaced with the information about the variants, vaccinations and decline in cases, which are explained well in the discussion part of the paper.
In introduction: Add a paragraph about different COVID-19 drugs starting from Plasmapheresis to Steroids, Anti-coagulants, Remdesivir given to the covid patients. Please consult following papers.
https://pubmed.ncbi.nlm.nih.gov/35585996/
https://pubmed.ncbi.nlm.nih.gov/34827332/
Table 1 shows role of comorbid. But comorbid role in not well discussed in the discussion part of the paper.
There is a high quality paper published recently on Global investments and pandemic preparedness and COVID-19 in Lancet group. Consult this paper and relate to your work in the discussion of paper.
https://www.thelancet.com/journals/langlo/article/PIIS2214-109X(23)00007-4/fulltext
Paragraph 297-110: ICU admission and deaths are discussed but the type of ventilation has also effect on patient outcome. Discuss, the role of invasive and non-invasive ventilation in the outcome patients in ICU.
Lines 312-315. Just the names of different type of drugs are written, which of them have a positive effect on patient survival, also discuss in this paragraph.
Conclusion of the paper is not well written. There are many unnecessary lines and important information missed e.g. delete lines 502-504. Write concisely the role of drugs, vaccines, variants in relation to different covid waves.
Author Response
Comment #1. Abstract: Add more results in abstract. Line 13-18 can be deleted and replaced with the information about the variants, vaccinations and decline in cases, which are explained well in the discussion part of the paper.
Response
The exclusion of certain data from the abstract of our scientific paper is due to two main reasons. Firstly, the abstract aims to provide a concise summary of the main ideas and findings of the study. In order to maintain focus on the central themes, it is necessary to exclude data that are not directly aligned with the core objectives and outcomes discussed in the abstract. However, these additional data are adequately addressed and discussed in detail throughout the body of the paper, ensuring their appropriate presentation and analysis.
Secondly, the specific journal to which we are submitting our paper has strict length restrictions for abstracts. These limitations are in place to ensure the concise nature of the abstract, enabling readers to quickly grasp the key aspects of the study. As a result, it becomes essential to prioritize the inclusion of essential information and avoid exceeding the allotted word count.
While it is acknowledged that the excluded data may contain valuable insights, their omission from the abstract does not diminish their significance within the overall context of the paper. The abstract serves as a succinct overview, guiding readers to delve deeper into the complete study for a comprehensive understanding of the research and the data presented.
Comment #2. In introduction: Add a paragraph about different COVID-19 drugs starting from Plasmapheresis to Steroids, Anti-coagulants, Remdesivir given to the covid patients. Please consult following papers.
https://pubmed.ncbi.nlm.nih.gov/35585996/
https://pubmed.ncbi.nlm.nih.gov/34827332/
Response
Although we believe it is out of the scope of the manuscript, we added:
“Several drugs have been introduced for the treatment of COVID-19. Corticosteroids, such as dexamethasone, have demonstrated efficacy in reducing inflammation and improving outcomes in severe cases. Plasmapheresis, a procedure that removes and replaces blood plasma, has been explored as a potential treatment option to remove harmful antibodies in critically ill patients. Anticoagulants, such as heparin, are administered to prevent blood clotting complications associated with COVID-19. Immunomodulators, such as tocilizumab, act to regulate the immune response and are utilized in severe cases with cytokine release syndrome. Antiviral drugs, including remdesivir, target the replication of the SARS-CoV-2 virus. These drugs, used in various combinations and based on disease severity, have shown promise in improving outcomes and reducing the severity of COVID-19.”
We also have included those two references.
Comment #3. Table 1 shows role of comorbid. But comorbid role in not well discussed in the discussion part of the paper.
Response
We have just added the following sentences:
“Our cohort had prevalent conditions such as type 2 diabetes, metabolic syndrome, or cardiovascular disease. These conditions, along with advanced age, have been associated with worse outcomes in individuals infected with SARS-CoV-2. Older patients with pre-existing conditions are particularly vulnerable, as age can weaken the immune system and make individuals more susceptible to severe illness. Moreover, comorbidities can further increase the severity of COVID-19 symptoms, contribute to a higher risk of complications, and lead to a higher mortality rate. These underlying health conditions and the aging process can exacerbate the inflammatory response triggered by the virus, resulting in complications such as acute respiratory distress syndrome and multiorgan dysfunction, which can explains the high mortality among the elderly in our cohort.”
Comment #4. There is a high quality paper published recently on Global investments and pandemic preparedness and COVID-19 in Lancet group. Consult this paper and relate to your work in the discussion of paper.
https://www.thelancet.com/journals/langlo/article/PIIS2214-109X(23)00007-4/fulltext
Response
We have added the following sentences:
“Finally, an intriguing study highlights a recurring pattern of panic and neglect in funding pandemic preparedness. Resources tend to increase in the aftermath of crises but subsequently decline. The high economic costs incurred by the COVID-19 pandemic further highlight the urgent need for investment in preparedness. Estimates for the annual funding required for pandemic preparedness vary, but they remain relatively small compared to the projected costs associated with events like COVID-19. Sustainable funding for pandemic preparedness necessitates effective collaboration between global health stakeholders and national health system leaders, as demonstrated by the importance of timely health responses when political commitment is present.”
Comment #5. Paragraph 297-310: ICU admission and deaths are discussed but the type of ventilation has also effect on patient outcome. Discuss, the role of invasive and non-invasive ventilation in the outcome patients in ICU.
Response
We have added a new paragraph, as follows:
“The choice between invasive and non-invasive ventilation plays a critical role in the outcome of patients with COVID-19 in ICUs. It is typically employed in patients with severe respiratory failure or acute respiratory distress syndrome (ARDS). In contrast, non-invasive ventilation provides respiratory support through a mask or nasal prongs without the need for intubation. While both ventilation strategies aim to support breathing, invasive ventilation is associated with higher levels of respiratory support and is often used in more critically ill patients. The choice of ventilation mode can significantly impact patient outcomes, with invasive ventilation generally being associated with higher mortality rates compared to non-invasive ventilation. However, the decision regarding the appropriate ventilation strategy should be individualized, taking into account factors such as disease severity, patient characteristics, and careful assessment of risks and benefits.”
Comment #6. Lines 312-315. Just the names of different type of drugs are written, which of them have a positive effect on patient survival, also discuss in this paragraph.
Response
We have added some sentences:
Corticosteroids are beneficial in treating severe COVID-19 cases by reducing lung inflammation and preventing complications. Dexamethasone, studied in the RECOVERY trial, has shown to reduce mortality in hospitalized patients requiring supplemental oxygen or mechanical ventilation. Remdesivir can shorten recovery time, especially in severe cases, but its impact on mortality reduction remains inconclusive. Combining immunomodulatory drug baricitinib with remdesivir has demonstrated faster recovery and improved outcomes in hospitalized patients, particularly those needing supplemental oxygen or high-flow therapy.
Comment #7. Conclusion of the paper is not well written. There are many unnecessary lines and important information missed e.g. delete lines 502-504. Write concisely the role of drugs, vaccines, variants in relation to different covid waves.
Response
We have rephrased all sentences to match the reviewer’s suggestions, as follows:
“In this study, our objective was to present a comprehensive overview of trends and patterns observed during various stages of the COVID-19 pandemic. We aimed to highlight differences in demographic data, clinical information, and healthcare indicators. Throughout the course of the pandemic, we witnessed advancements in patient management, such as the development of new drugs and the rollout of vaccination programs. However, we also observed the emergence of different variants and lineages, which ultimately had a significant impact on hospitalization rates and mortality trends. When formulating nonmedical strategies to address future waves of COVID-19 or outbreaks of new infectious diseases, it is crucial to consider these factors, in addition to the clinical experience gained during the pandemic. To gain insights into the diverse patient profiles, we analyzed administrative and demographic data across different waves of the pandemic. By dividing the pandemic into distinct waves, we were able to identify variations in patient demographics, likely influenced by key milestones preceding each wave. These milestones included factors such as the initiation of vaccination programs, changes in SARS-CoV-2 variants, and the implementation of social distancing measures. Our findings suggest that analyzing the pandemic as a whole may not capture the complete picture, and it is more informative to examine individual waves. Factors specific to each wave, such as the timing of COVID-19 infections, vaccination rates, and predominant virus variants, should be considered when designing future clinical studies on the pandemic. As a result, our research provides a general analytical framework that can be applied to other settings.”
Round 2
Reviewer 3 Report
Reviewer comments are well addressed by the authors. Manuscript is much improved and Acceptable in present form.